# Evaluating the Usefulness of the Blood Apolipoprotein A2 Isoform Index for Pancreatic Cancer Diagnosis

**DOI:** 10.3390/cancers17071071

**Published:** 2025-03-22

**Authors:** Kento Shionoya, Atsushi Sofuni, Shuntaro Mukai, Takayoshi Tsuchiya, Reina Tanaka, Ryosuke Tonozuka, Kenjiro Yamamoto, Kazumasa Nagai, Yukitoshi Matsunami, Hiroyuki Kojima, Hirohito Minami, Noriyuki Hirakawa, Kyoko Asano, Yuma Yamaguchi, Kazuki Hama, Takao Itoi

**Affiliations:** 1Department of Gastroenterology and Hepatology, Tokyo Medical University, Tokyo 160-0023, Japan; kjscf034@tokyo-med.ac.jp (K.S.); a-sofuni@tokyo-med.ac.jp (A.S.); s-mukai@tokyo-med.ac.jp (S.M.); tsuchiya@tokyo-med.ac.jp (T.T.); r-tanaka@tokyo-med.ac.jp (R.T.); tonozuka@tokyo-med.ac.jp (R.T.); kenjiro@tokyo-med.ac.jp (K.Y.); kazu4439@tokyo-med.ac.jp (K.N.); ym1228@tokyo-med.ac.jp (Y.M.); kojima-h@tokyo-med.ac.jp (H.K.); minami@tokyo-med.ac.jp (H.M.); d121037@tokyo-med.ac.jp (N.H.); kyoko_29@tokyo-med.ac.jp (K.A.); yumayama@tokyo-med.ac.jp (Y.Y.); k-hama@tokyo-med.ac.jp (K.H.); 2Department of Clinical Oncology, Tokyo Medical University, Tokyo 160-0023, Japan

**Keywords:** apolipoprotein A2, carbohydrate antigen 19-9, pancreatic neoplasms, tumor markers, diagnostic tests, early detection, endoscopic ultrasound, surveillance, diagnostic accuracy

## Abstract

This study evaluated the diagnostic effectiveness of the Apolipoprotein A2-isoform (APOA2-i) Index for early detection of pancreatic cancer in combination with the conventional tumor marker CA 19-9. In a cohort of patients, APOA2-i demonstrated lower sensitivity and specificity for advanced pancreatic cancer stages (II–IV) compared to CA 19-9. However, it outperformed CA 19-9 in identifying early-stage cancers (stages 0 and I), successfully detecting cancers that were CA 19-9-negative. The combination of both biomarkers showed improved diagnostic accuracy for early-stage pancreatic cancer, highlighting the potential of APOA2-i as a valuable tool in clinical settings. The study’s findings suggest that integrating the APOA2-i Index with CA 19-9 could significantly enhance early detection and surveillance in high-risk patients, ultimately contributing to better clinical outcomes and treatment strategies for pancreatic cancer.

## 1. Introduction

Pancreatic cancer (PC) is a well-known disease with poor prognosis and an overall five-year survival rate of approximately 10%. Diagnosis is often delayed until symptoms, such as jaundice, weight loss, or appetite loss, emerge, which complicates early intervention. Although early-stage diagnoses and advancements in chemotherapy and surgery have improved prognosis, PC still has a higher mortality rate than other cancers [1,2,3,4]. More than 70% of PC cases are already unresectable at diagnosis [5]; hence, early detection is important to improve the survival rate and prognosis of patients with PC.

Carbohydrate antigen 19-9 (CA 19-9) is the most widely used blood-based biomarker for PC; however, it has limitations in detecting early-stage PC and can also be elevated in other malignant diseases, such as biliary tract and colorectal cancers [6,7,8,9]. Approximately 50% of PC patients with tumors < 3 cm in size do not show elevated serum CA 19-9 levels [10,11], and up to 10% of patients with PC and Lewis antigen-negative blood types, which cannot be measured in daily clinical practice, are unable to synthesize CA 19-9, resulting in false-negative results [12,13].

Honda et al. reported that alterations in the processing patterns of C-terminal amino acids in circulating apolipoprotein A2 (APOA2)-homodimers, specifically APOA2-isoforms (APOA2-i), are seen in patients with PC and high-risk individuals for PC [14]. The APOA2-AT/TQ (APOA2-i Index) is significantly reduced in patients with PC due to pancreatic exocrine dysfunction causing aberrant processing of APOA2 dimers [15]. The APOA2-i Index was approved as a tumor marker for PC in Japan in March 2024 and is now utilized clinically; however, there is a lack of studies evaluating its effectiveness in clinical practice. This study aimed to evaluate the clinical utility of the APOA2-i Index (Toray, Tokyo, Japan) in diagnosing PC.

## 2. Materials and Methods

### 2.1. Patients

This retrospective study was conducted at Tokyo Medical University Hospital, and analyzed the data of patients who underwent endoscopic ultrasound-guided tissue sampling between March 2024 and December 2024 at a single-center pancreatic disease care center in Japan.

The inclusion criteria were as follows: (1) 20 years and older; (2) PC was suspected on images or blood tests; and (3) patients underwent CA 19-9 and APOA2-i Index measurement. The exclusion criteria were as follows: (1) patients under 20 years old and (2) patients who expressed a desire to not participate in the study.

This study was reviewed and approved by our institutional review board (IRB no.: T2024-0074, 29 January 2025), and informed consent was obtained from all participants using the opt-out method on our institution website and in-hospital posting, as this was a retrospective study that utilized data from medical charts and computerized records. This study was conducted in accordance with the ethical standards of the 1964 Declaration of Helsinki and its later amendments.

### 2.2. Outcomes and Definitions

This study aimed to clarify the diagnostic performance of the APOA2-I Index, a novel biomarker for PC. The study outcomes included the sensitivity, specificity, positive predictive value (PPV), and negative predictive value (NPV) of the APOA2-i Index and CA 19-9 for PC. Sensitivity was defined as the number of true positives/the number of true positives + the number of false negatives. Specificity was defined as the number of true negatives/the number of true negatives + the number of false negatives. PPV was defined as the number of true positives/the number of true positives + the number of false positives. NPV was defined as true negatives/the number of true negatives + the number of false negatives.

The concentrations of APOA2-AT/AT and APOA2-TQ/TQ were determined using the APOA2-AT and APOA2-TQ values, respectively, as measured by in vitro diagnostic enzyme-linked immunosorbent assay. APOA2-i Index was defined as the concentration of APOA2-AT/TQ, which was calculated using the formula:√(APOA2 _AT/TQ_) = (APOA2_AT_ × APOA2_TQ_)(1)

The cut-off value for a positive APOA2-i Index was defined as <59.5 μg/mL, as used in clinical settings [15]. The concentration of CA 19-9 was measured using a chemiluminescent enzyme immunoassay kit for CA 19-9 (Lumipulse Presto CA 19-9; FUJIREBIO INC., Tokyo, Japan). The cut-off value of CA 19-9 was prespecified as 37.0 U/mL, as used in clinical settings. PC stages were based on the TNM Classification of Malignant Tumors, eighth edition. Stage 0 or I was defined as early stage PC, and stages II–IV were defined as advanced stage [16].

### 2.3. Statistical Analyses

Continuous variables were presented as medians with interquartile ranges. Binary variables were compared using Fisher’s exact test, and continuous variables were compared using the Mann–Whitney U test, Kruskal–Wallis test, or multivariate analysis. Logistic regression analyses were performed to calculate the odds ratios (ORs) and 95% confidence intervals (CIs) for the diagnostic performance of the biomarkers. We inputted potential predictive factors, namely age, sex, and whether the patient had PC or CP, for the diagnosis of PC (univariate analysis *p* < 0.05) into a multivariate model. Statistical significance was set at *p* < 0.05. Receiver operating characteristic (ROC) curves were developed for the diagnosis rates. All statistical analyses were performed using SPSS version 29.0.1.0 (IBM Corp., Armonk, NY, USA).

## 3. Results

### 3.1. Patient Characteristics

Table 1 shows the patient characteristics. A total of 174 patients were included in this study, comprising 99 men and 75 women, with a median age of 70.5 years (range 25–94 years). We measured the serum levels of APOA2-i Index and CA 19-9 in 76 patients with PC diagnosed using endoscopic ultrasound (EUS)-guided tissue sampling (Stage 0, n = 5; I, n = 4; II, n = 15; III, n = 19; and IV, n = 33) and 98 patients with non-PC (intraductal papillary mucinous neoplasm [IPMN], n = 36; chronic pancreatitis [CP], n = 33; pancreatic neuroendocrine neoplasm, n = 8; autoimmune pancreatitis, n = 9; and others, n = 12).

### 3.2. Diagnostic Performance of APOA2-i Index

The diagnostic performance results of the APOA2-i Index are presented in Table 2, Table 3 and Table 4 in column A. Among 174 patients, 58 (33.3%) tested positive for the APOA2-i Index: 37/76 (48.7%) patients with PC and 21/98 (21.4%) patients with non-PC. The sensitivity, specificity, PPV and NPV of APOA2-i Index alone were as follows: all stages of PC, 48.7%, 78.6%, 63.8%, and 66.4%; stages 0 and I, 33.3%, 66.7%, 5.2%, and 94.8%; and stages II–IV, 50.7%, 77.6%, 58.6%, and 71.6%. The ROC curve yielded an area under the curve (AUC) of 0.697 (Figure 1a). By setting the cut-off for the APOA2-i Index to 56.8, the AUC was maximized to 0.707 (Figure 1b). For patients with non-PC, 24.2% (8/33) and 16.7% (6/36) of patients with CP and IPMN, respectively, were positive for the APOA2-i Index.

The univariate analysis identified PC as a risk factor for APOA2-i Index biomarker positivity (*p* < 0.001). Furthermore, the multivariate analysis identified PC as an independent risk factor for APOA2-i Index biomarker positivity (OR: 3.48, *p* < 0.001).

### 3.3. Diagnostic Performance of CA 19-9

The diagnostic performance results of CA 19-9 are presented in Table 2, Table 3 and Table 4. Among 174 patients, 69 (39.7%) tested positive for CA 19-9: 58/76 (76.3%) patients with PC and 11/98 (11.2%) patients with non-PC. The sensitivity, specificity, PPV and NPV of CA 19-9 alone were as follows: all stages of PC, 76.3%, 88.8%, 84.1%, and 82.9%; stages 0 and I, 22.2%, 59.4%, 2.9%, and 93.3%; and stages II–IV, 83.6%, 87.9%, 81.2%, and 89.5%. The ROC curve yielded an AUC of 0.785 (Figure 2). For patients with non-PC, 12.1% (4/33) and 5.6% (2/36) of patients with CP and IPMN, respectively, were positive for CA 19-9.

The univariate analysis identified PC (*p* < 0.001) and CP (*p* < 0.001) as risk factors for CA 19-9 biomarker positivity. Furthermore, the multivariate analysis identified PC as an independent risk factor for CA 19-9 biomarker positivity (OR: 25.5, *p* < 0.001).

### 3.4. Diagnostic Performance of Combination of APOA2-i Index with CA 19-9

The diagnostic performance results for the combination of APOA2-i Index and CA 19-9 are presented in Table 2, Table 3 and Table 4. Among the 174 patients, 92 (52.9%) tested positive for either APOA2-i Index or CA 19-9: 64/76 (84.2%) patients with PC and 28/98 (28.6%) patients with non-PC. The sensitivity, specificity, PPV and NPV for the combination of APOA2-i Index and CA 19-9 were as follows: all stages of PC, 82.1% and 70.8%; stages 0 and I, 44.4% and 46.7%; and stages II–IV, 89.6% and 70.1%. For patients with non-PC, 30.3% (10/23) and 22.2% (8/36) of patients with CP and IPMN, respectively, were positive for one of the tests. Two of six (33.3%) healthy subjects tested positive for one of the tests. One of the patients that was originally followed up for CP tested positive for both APOA2-i Index and CA 19-9. Further imaging studies such as EUS and CT led to the diagnosis of PC. The univariate analysis identified PC as a risk factor for both APOA2-i Index and CA 19-9 biomarker positivity (*p* < 0.001). Furthermore, the multivariate analysis identified PC as an independent risk factor for both APOA2-i Index and CA 19-9 biomarker positivity (OR: 13.3, *p* < 0.001).

The APOA2-i Index showed lower accuracy for advanced PC cases compared to CA 19-9, but it provided superior accuracy for early stage PC detection. Three early stage PC cases negative for CA 19-9 were detected with the APOA2-i Index, demonstrating high diagnostic accuracy for early stage PC when both biomarkers are combined.

## 4. Discussion

In recent years, the development of surgery and chemotherapy has improved the prognosis of PC; however, PC remains a disease with a poor prognosis [1,2,3,4]. Comprehensive genomic profiling contributes to the improvement of prognosis in other cancers; however, it has not yet sufficiently contributed to improving the prognosis of PC because therapeutic target gene mutations are rarely found and few therapeutic agents are available [17,18]. Early diagnosis and timely treatment are critical for PC. Lesions in the head of the pancreas may be detected relatively early because of obstructive jaundice or acute cholangitis, but lesions in the body or tail are often asymptomatic, making early diagnosis difficult. Approximately 10–15% of patients with PC are diagnosed with resectable or borderline resectable disease [19]. Computed tomography (CT) scans are generally performed as an initial examination. In previous reports, the sensitivity, specificity, and diagnostic rate of CT were 94.1–97%, 80–83%, and 90.4–96%, respectively. The sensitivity and specificity for the detection of small pancreatic masses was 77% and 100%, respectively [3,20,21,22,23,24,25,26,27]. EUS is also useful in diagnosing PC, with a 96.9–100% presence rate and an 82.6–96% qualitative rate, which is higher than that of CT [28,29,30,31,32,33,34]. Identifying high-risk cases of PC at an early stage using minimally invasive tests, such as blood and urine tests, is crucial. These tests should be followed by imaging and histological evaluation using EUS-guided tissue sampling to diagnose, treat, and improve the prognosis of early stage PC.

The use of serum CA 19-9 as a biomarker for PC has limitations such as the failure to yield positive results in small lesions, difficulty of early tumor detection, and positive results for other cancers [7,8,9,10,35]. To enable early diagnosis of PC, saliva or urine tests and methods to measure the amount of circular RNA in the blood have been developed; however, only a few of these methods have been actively applied in clinical practice [35,36]. A simple and accurate measurement method has long been desired. In 2024, the novel biomarker kit APOA2-i Index for PC diagnosis was released in Japan. The APOA2-i Index is primarily reflective of pancreatic exocrine dysfunction caused by PC, which serves as a marker for early detection. The mechanism underlying this index involves three primary isoforms of APOA2 that exist in the bloodstream as dimers: APOA2-TQ/TQ, APOA2-TQ/AT, and APOA2-AT/AT. The differences in amino acids at the C-terminus of each isoform are believed to arise from cleavage of the C-terminus by pancreatic-derived exopeptidases, such as carboxypeptidase A [14,15]. A previous study reported alterations in the processing patterns of C-terminal amino acids of circulating APOA2 homodimer in patients with PC and in high-risk individuals for PC [14]. The APOA2-i Index is significantly reduced in patients with PC because the alteration in pancreatic exocrine functions causes aberrant processing of APOA2 dimers [15]. In the case of PC, the cleavage pattern of the C-terminal amino acids undergoes a transformation, leading to increased cleavage and resulting in the predominance of the APOA2-AT/AT isoform. Conversely, there is also a cleavage suppression pattern, leading to a predominance of the APOA2-TQ/TQ isoform. In both cleavage patterns, the amount of the intermediate cleavage product, APOA2-AT/TQ, decreases. The APOA2-i Index is calculated by measuring the concentrations of APOA2-AT and APOA2-TQ in the plasma or serum and determining their geometric mean, which is defined as the concentration of APOA2-AT multiplied by the concentration of APOA2-TQ. The APOA2-i Index has been shown to correlate with the concentrations of APOA2-AT/TQ in the bloodstream and reflects a decline in exocrine function due to PC. During the development phase, the diagnostic rate for PC was not explored for stages III and IV; however, for stages I and II, it had a 60.2% rate with a specificity of 55.3% (26/47). The point estimate of the ROC-AUC for APOA2-i Index was 0.836 when the cutoff value was 54.47 μg/mL (95% CI: 0.774–0.898), while the point estimate of the ROC-AUC for serum CA 19-9 was 0.783 (95% CI: 0.710–0.855). The difference between the ROC-AUC for the APOA2-i Index and CA 19-9 was 0.053. In a previous report, even among healthy individuals, a 4.9% positivity rate of APOA2-i Index was observed; although the number of cases was small, 16.6% (1/6) of healthy subjects were positive in the current study [15]. The overall positivity rates of the APOA2-i Index in clinical practice were 33.3% (58/174) in total and 48.7% (37/76) in early stages for patients with PC in this study.

Along with PC, CP is another condition that reflects pancreatic exocrine dysfunction. When CP is present, the baseline pancreatic exocrine function is already compromised, which could lead to a positive APOA2-i Index result, even in the absence of PC. In our study, although CP did not emerge as an independent factor for diagnostic performance, 24% of cases (8/33) with CP tested positive, indicating a noteworthy proportion of false positives. The univariate analysis identified CP as a factor associated with positive CA 19-9 results; however, it was not associated with the APOA2-i Index. In one patient who was followed up for CP, a positive APOA2-i Index and CA 19-9 led to a diagnosis of PC, which was considered valid. In cases wherein false positive results are suspected, a combination of blood tests and imaging studies should be performed at short intervals, and follow-up should be performed for malignant findings.

IPMN is also known to be a risk factor for PC, and high serum CA 19-9 levels are considered a relative risk factor for the malignant transformation of PC and IPMN; hence, the measurement of serum CA 19-9 level is recommended in the guidelines for managing cases of IPMN. Patients with IPMN often have diabetes mellitus and impaired pancreatic exocrine function, with marked atrophy of the pancreatic parenchyma due to mucus filling the pancreatic duct [37,38]. IPMN also results in impaired exocrine function, which may lead to APOA2-i Index positivity. In this study, some IPMN cases were positive, but no cases of IPMN with PC were diagnosed by imaging that was triggered by a positive APOA2-i Index. Since this is a study with a small number of cases, it is necessary to examine its usefulness as a marker for IPMN with PC.

The multivariate analysis revealed that only the presence of PC was an independent factor for APOA2-i Index positivity, suggesting its effectiveness as a PC marker. Although this biomarker was positive, which reflects reduced pancreatic exocrine function, no statistically significant difference was observed between the patients with and without diabetes mellitus. This may be because type II diabetes is due to impaired glucose tolerance and pancreatic diabetes is due to impaired insulin secretion caused by endocrine dysfunction, and the present marker reflects impaired exocrine function, which is a different mechanism. The mechanism of action of this biomarker may be applicable for the early detection of CP, a risk factor for PC. Cases exhibiting exocrine dysfunction due to a tumor causing stenosis of the main pancreatic duct (MPD) may yield positive results relatively early. In contrast, tumors located more distally from the MPD may present challenges for early diagnosis. Therefore, the accumulation of additional cases is necessary for future studies.

Regarding standalone diagnostic performance, CA 19-9 was superior to the APOA2-i Index. Notably, among the seven early stage PC cases that were negative for CA 19-9, three had positive APOA2-i Index results. There were seven cases that were positive for both CA 19-9 and APOA2-i Index. Three out of the seven cases had early stage PC, while the remaining four cases had advanced PC. The reason the APOA2-i index has higher sensitivity and specificity than CA19-9 in early stages is not that it diagnoses early stage cancer, but that CA19-9 is unable to sufficiently detect the functional decline in exocrine pancreatic function seen in early stages, whereas the APOA2-i index is able to detect this decline more sensitively. The APOA2-i Index is considered positive below 59.5 μg/mL (ROC-AUC: 0.697); however, clinically, a cutoff of 56.8 μg/mL (ROC-AUC: 0.707) provided better diagnostic capability for detecting PC. The diagnostic rate of PC as a stand-alone biomarker is not satisfactory; hence, better cutoff values are necessary for the diagnosis of PC. Combining CA 19-9 with the APOA2-i Index resulted in a diagnostic rate of 82.1%, enhancing the potential for early stage PC detection. Although it is a pancreas-specific marker, another issue is that the APOA2-i Index is somewhat more expensive to measure than CA 19-9. Measuring CA19-9 takes about an hour to measure and costs about $8, while the APOA2-i Index takes about one week to measure and costs about $20.

Despite the relevant findings, this study has several limitations, including its retrospective nature, single-center focus, and relatively small sample size. Further studies addressing these limitations should be performed to validate our findings.

## 5. Conclusions

Although APOA2 is inferior to CA19-9 in terms of PC diagnostic ability alone, the APOA2-i Index combined with CA 19-9 may improve early stage PC detection, especially in diagnostically challenging cases and high-risk patient surveillance.

## Figures and Tables

**Figure 1 cancers-17-01071-f001:**
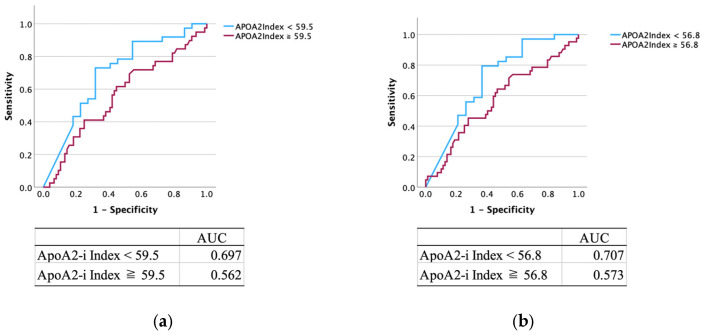
ROC curve of the APOA2-i Index. The ROC curve yielded an area under the curve of (**a**) 0.697 and (**b**) 0.707 when the cutoff values for the APOA2-i Index were set at 59.5 μg/mL and 56.8 μg/mL, respectively. ROC, receiver operating characteristic; APOA2-i, apolipoprotein A2-isoform; AUC, area under the curve.

**Figure 2 cancers-17-01071-f002:**
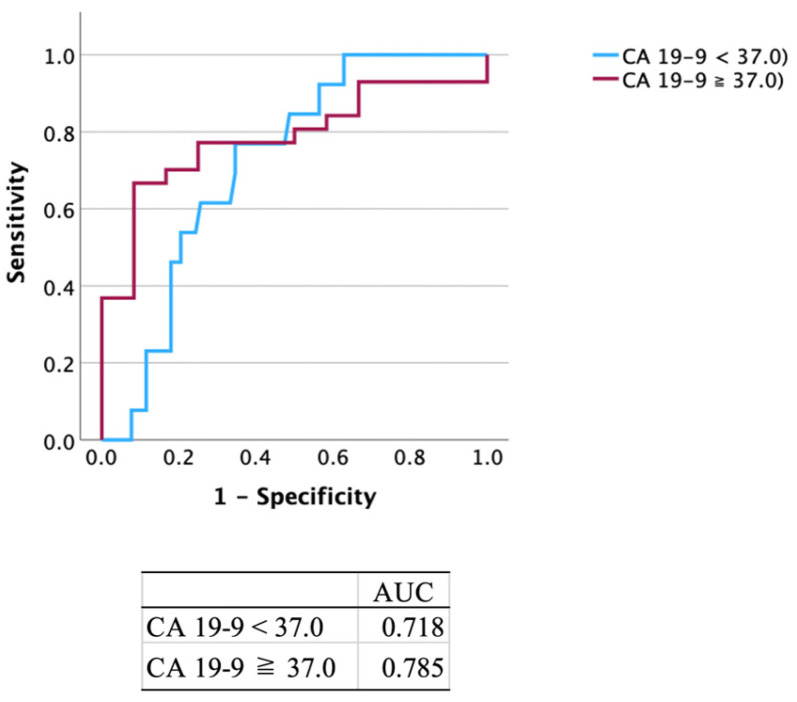
ROC curve of CA 19-9. The ROC curve yielded an area under the curve of 0.785 when the cutoff value for the CA 19-9 was set at 37.0 μg/mL. ROC, receiver operating characteristic; CA 19-9, carbohydrate antigen 19-9; AUC, area under the curve.

**Table 1 cancers-17-01071-t001:** Patient characteristics.

Age (Year)	Median (Year) [Range]	70.5 [25–94]
Sex	Male	99
	Female	75
Diseases	PC	76
	Intraductal papillary mucinous neoplasm	36
	Chronic pancreatitis	33
	Auto immune pancreatitis	9
	Pancreatic neuroendocrine neoplasm	8
	Others	12
PC stages	Stage 0	5
	Stage I	4
	Stage II	15
	Stage III	19
	Stage IV	33
Biomarker	ApoA2-I Index positive	58
	CA 19-9 positive	69
	Either positive	92

CA 19-9, Carbohydrate antigen 19-9; PC, pancreatic cancer.

**Table 2 cancers-17-01071-t002:** Diagnostic results for the two types of biomarkers.

		ApoA2-i Index	CA 19-9	Either Positive
Overall results	33.3% (58/174)	33.3% (58/174)	39.7% (69/174)	52.9% (92/174)
Age	65 year >	28.8% (19/66)	34.8% (23/66)	51.5% (34/66)
	65 year ≦	36.1% (39/108)	39.8% (43/108)	53.7% (58/108)
	*p*-value	0.32	0.96	0.78
Sex	Male	37.4% (37/99)	35.4% (35/99)	49.5% (49/99)
	Female	28% (21/75)	45.3% (34/75)	60% (45/75)
	*p*-value	0.18	0.20	0.31
PC	PC cases	48.7% (37/76)	76.3% (58/76)	84.2% (64/76)
	non PC cases	21.4% (21/98)	11.2% (11/98)	28.6% (28/98)
	*p*-value	<0.001	<0.001	<0.001
CP	CP cases	24.2% (8/33)	12.1% (4/33)	30.3% (10/33)
	non CP cases	35.5% (50/141)	46.1% (65/141)	58.2% (82/141)
	*p*-value	0.22	<0.001	0.004
PC stage	Stage 0 or I	33.3% (3/9)	22.2% (2/9)	44.4% (4/9)
	Others	33.3% (55/165)	40.6% (67/165)	53.3% (88/165)
	*p*-value	N/A	0.27	0.60
	Stage II to IV	50.7% (34/67)	83.6% (56/67)	89.6% (60/67)
	Others	22.4% (24/107)	12.1% (13/107)	29.9% (32/107)
	*p*-value	<0.001	<0.001	<0.001

CA 19-9, Carbohydrate antigen 19-9; CP, chronic pancreatitis; N/A, not applicable; PC, pancreatic cancer.

**Table 3 cancers-17-01071-t003:** Comparison of diagnostic results between the three groups.

		Sensitivity	Specificity	PPV	NPV
ApoA2-i Index	PC	48.7% (37/76)	78.6% (77/98)	63.8% (37/58)	66.4% (77/116)
	Stage 0 or I	33.3% (3/9)	66.7% (110/165)	5.2% (3/58)	94.8% (110/116)
	Stage II to IV	50.7% (34/67)	77.6% (83/107)	58.6% (34/58)	71.6% (83/116)
CA 19-9	PC	76.3% (58/76)	88.8% (87/98)	84.1% (58/69)	82.9% (87/105)
	Stage 0 or I	22.2% (2/9)	59.4% (98/165)	2.9% (2/69)	93.3% (98/105)
	Stage II to IV	83.6% (56/67)	87.9% (94/107)	81.2% (56/69)	89.5% (94/105)
Either positive	PC	84.2% (62/76)	71.4% (70/98)	69.6% (64/92)	85.4% (70/82)
	Stage 0 or I	44.4% (4/9)	46.7% (77/165)	4.3% (4/92)	93.9% (77/82)
	Stage II to IV	89.6% (60/67)	70.1% (75/107)	65.2% (60/92)	91.5% (75/82)

CA 19-9, Carbohydrate antigen 19-9; CP, chronic pancreatitis; NPV, negative predictive value; PPV, positive predictive value.

**Table 4 cancers-17-01071-t004:** Statistical analysis of diagnostic performance.

	Univariate Analysis	Multivariate Analysis		
ApoA2-i Index	*p*-Value	OR	95% CI	*p*-Value
PC cases	<0.001	3.48	1.80–6.73	<0.001
Age	0.50			
Sex	0.17			
CP cases	0.44			
**CA 19-9**				
PC cases	<0.001	25.5	11.2–57.9	<0.001
Age	0.33			
Sex	0.088			
CP cases	0.84			
**Either positive**				
PC cases	<0.001	13.3	6.26–28.4	<0.001
Age	0.64			
Sex	0.24			
CP cases	0.79			

CA 19-9, Carbohydrate antigen 19-9; CP, chronic pancreatitis; PC, pancreatic cancer.

## Data Availability

The data presented in this study are available on request from the corresponding author. The data are not publicly available due to the ethics approval agreement.

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
