# Peer review of "Evaluating the Usefulness of the Blood Apolipoprotein A2 Isoform Index for Pancreatic Cancer Diagnosis"

_cancers, 2025, doi:10.3390/cancers17071071_

Round 1
Reviewer 1 Report
Comments and Suggestions for Authors
The early detection of pancreatic cancer proves difficult with existing tumor markers such as carbohydrate antigen 19-9 (CA 19-9), highlighting the need for novel biomarkers. Apolipoprotein A2 (APOA2) is not directly produced by the tumor, but its distinct mechanisms may aid in detecting pancreatic cancer compared to CA 19-9. This study evaluated the diagnostic performance of the APOA2 isoform (APOA2-i) index among patients with pancreatic cancer. The APOA2-i index levels and CA 19-9 were measured in the serum of 76 patients diagnosed with pancreatic cancer and 98 without to assess diagnostic efficacy. The results indicated that while the APOA2-i index demonstrated less precision than CA 19-9 in detecting advanced-stage (stages II–IV) pancreatic cancer, it exhibited superior precision for early-stage (stages I and 0) detection. Notably, the APOA2-i index successfully identified two out of three early-stage pancreatic cancer cases that tested negative for CA 19-9, suggesting a high diagnostic precision when combining the two biomarkers for early-stage pancreatic cancer. Multivariable analysis established pancreatic cancer as an independent risk factor for a positive APOA2-i index, as well as for CA 19-9. Thus, the combined use of the APOA2-i index and CA 19-9 could enhance the detection of early-stage pancreatic cancer, particularly in challenging cases and for the surveillance of high-risk patients. It is recommended that this manuscript be accepted after revision, as suggested below:
- It is recommended that the authors highlight the tumor heterogeneity and tumor microenvironment specific to pancreatic cancer in the introduction section, as these characteristics are pivotal to understanding this malignancy. The following references are suggested to be cited for an in-depth discussion of this aspect.
[1] A. Gu, J. Li, M.-Y. Li, Y. Liu, Patient-derived xenograft model in cancer: establishment and applications. MedComm, 2025, 6, e70059. DOI: 10.1002/mco2.70059
[2] Zhong J, Shi J, Amundadottir L T. Artificial intelligence and improved early detection for pancreatic cancer. The Innovation, 2023, 4(4), 100457. https://doi.org/10.1016/j.xinn.2023.100457
- The author should present innovative treatment modalities for pancreatic cancer, such as nanotherapy, immunotherapy, and cell-based therapies. While traditional surgical and pharmacological treatments have been foundational, they are now considered relatively outdated. It is recommended to cite the following literature. (Wang J, Liao Z-X. Research progress of microrobots in tumor drug delivery. Food & Medicine Homology, 2024, 1(2): 9420025. https://doi.org/10.26599/FMH.2024.9420025)
- The expression of the APOA2-i Index should be analyzed in other forms of cancer, such as biliary tract and colorectal cancer, to ascertain its specificity. This will elucidate the applicable scope of the APOA2-i Index as a marker for pancreatic cancer.
- Given the positive findings associated with the APOA2-i Index in patients with non-pancreatic cancer, it is suggested that further comprehensive clinical analyses be undertaken to discern potential causative factors.
- Given the cost and time limitations associated with the APOA2-i Index test, it is advisable for the authors to explore strategies to mitigate these challenges in clinical settings. This could include investigating more efficient testing methodologies or pioneering novel detection technologies.
Author Response
Comments 1:
It is recommended that the authors highlight the tumor heterogeneity and tumor microenvironment specific to pancreatic cancer in the introduction section, as these characteristics are pivotal to understanding this malignancy. The following references are suggested to be cited for an in-depth discussion of this aspect.
[1] A. Gu, J. Li, M.-Y. Li, Y. Liu, Patient-derived xenograft model in cancer: establishment and applications. MedComm, 2025, 6, e70059. DOI: 10.1002/mco2.70059
[2] Zhong J, Shi J, Amundadottir L T. Artificial intelligence and improved early detection for pancreatic cancer. The Innovation, 2023, 4(4), 100457. https://doi.org/10.1016/j.xinn.2023.100457
Response 1:
Thank you for your recommendation. We cited these [1] articles.
Comments 2:
The author should present innovative treatment modalities for pancreatic cancer, such as nanotherapy, immunotherapy, and cell-based therapies. While traditional surgical and pharmacological treatments have been foundational, they are now considered relatively outdated. It is recommended to cite the following literature. (Wang J, Liao Z-X. Research progress of microrobots in tumor drug delivery. Food & Medicine Homology, 2024, 1(2): 9420025. https://doi.org/10.26599/FMH.2024.9420025)
Response 2: Thank you for your recommendation. We do not cite it because we considered the content is somewhat out of line. We apologize.
Comments 3:
The expression of the APOA2-i Index should be analyzed in other forms of cancer, such as biliary tract and colorectal cancer, to ascertain its specificity. This will elucidate the applicable scope of the APOA2-i Index as a marker for pancreatic cancer.
Response 3:Thank you for your comment. We also considered APOA2-i Index should be analyzed in other forms of cancer which CA 19-9 may arise. However, now Japanese insurance system does not cover with other cancers. In addition, this study is retrospective study, now we cannot check it.
Comments 4:
Given the positive findings associated with the APOA2-i Index in patients with non-pancreatic cancer, it is suggested that further comprehensive clinical analyses be undertaken to discern potential causative factors.
Response 4:
Thank you for your important comment. We have already discussed in the text the possibility that it can be raised in chronic pancreatitis and IPMN. Since diabetes may develop or worsen with pancreatic cancer, we also examined the presence or absence of diabetes, but this was considered to be a negative factor for the marker to be elevated. This may be because type â…¡ diabetes is due to impaired glucose tolerance and pancreatic diabetes is due to impaired insulin secretion caused by endocrine dysfunction, and the present marker reflects impaired exocrine function, which is a different mechanism. We have added this point.
Comments 5:
Given the cost and time limitations associated with the APOA2-i Index test, it is advisable for the authors to explore strategies to mitigate these challenges in clinical settings. This could include investigating more efficient testing methodologies or pioneering novel detection technologies.
Response 5:
Thank you for your comments. In Japan, measuring CA19-9 takes about an hour to measure and costs about $8, while the ApoA2-i Index takes about one week to measure and costs about $20. We have added this point. (p9 l 302-304)
Reviewer 2 Report
Comments and Suggestions for Authors
This manuscript compares the usefulness of the Apolipoprotein A2 isomer index (APOA2-i index) as a biomarker for diagnosing pancreatic cancer with CA19-9 in 174 patients who were suspected of having pancreatic cancer and underwent endoscopic ultrasonography. 76 patients were diagnosed with pancreatic cancer, with 5 cases of stage 0, 4 cases of stage I, 15 cases of stage II, 19 cases of stage III, stage IV was 33 cases. Although there have been reports of papers that have investigated the usefulness of the APOA2-i index in diagnosing pancreatic cancer in large-scale studies, it is thought that there is great clinical significance in examining the usefulness of the APOA2-i index in a single facility with a well-organized clinical background. However, there are problems as shown below, and the authors should change the way they describe the paper.
- Overall, CA19-9 is a better biomarker for diagnosing pancreatic cancer, so this should be emphasized in the conclusion.
- I think it is more accurate to say that the reason the APOA2-i index has higher sensitivity and specificity than CA19-9 in Stages 0 and I is not that it diagnoses early-stage cancer, but that CA19-9 is unable to sufficiently detect the functional decline in exocrine pancreatic function seen in Stages 0 and I, whereas the APOA2-i index is able to detect this decline more sensitively. As stated in the main text, the APOA2-i index is not used to diagnose pancreatic cancer, but rather as a biomarker to detect precancerous lesions (similar to liver cirrhosis in hepatocellular carcinoma).
- For the reasons given above, there are still some questions about whether the diagnostic ability for pancreatic cancer can be improved by combining the APOA2-i index and CA19-9, and it seems that the APOA2-i index is only good at identifying high-risk groups for pancreatic cancer.
- The discussion is too long, so it would be better to focus on clearly showing the differences from previous large-scale studies.
Author Response
Comments 1: Overall, CA19-9 is a better biomarker for diagnosing pancreatic cancer, so this should be emphasized in the conclusion.
Response 1:
Thank you for your comment. We also considered CA 19-9 is better biomarker. We added in the conclusion section. (p9 l 307-308)
Comments 2:
I think it is more accurate to say that the reason the APOA2-i index has higher sensitivity and specificity than CA19-9 in Stages 0 and I is not that it diagnoses early-stage cancer, but that CA19-9 is unable to sufficiently detect the functional decline in exocrine pancreatic function seen in Stages 0 and I, whereas the APOA2-i index is able to detect this decline more sensitively. As stated in the main text, the APOA2-i index is not used to diagnose pancreatic cancer, but rather as a biomarker to detect precancerous lesions (similar to liver cirrhosis in hepatocellular carcinoma).
Response 2:
Thank you for your important comment. I think your comment is indeed an accurate way to phrase it as you have pointed out. We have added it. (p9 l290-294)
Comments 3:
For the reasons given above, there are still some questions about whether the diagnostic ability for pancreatic cancer can be improved by combining the APOA2-i index and CA19-9, and it seems that the APOA2-i index is only good at identifying high-risk groups for pancreatic cancer.
Response 3:
Thank you for your comment. As you have pointed out, even in the current insurance system in Japan, the use of the APOA2-i index is limited to patients who are considered to be at high risk for pancreatic cancer, and there are only a few hospitals where the test can be performed, and the insurance system is very restrictive. Therefore, CA 19-9 remains the best marker for the diagnosis of pancreatic cancer, but it can complement CA 19-9 measurement by detecting early-stage pancreatic exocrine hypofunction that cannot be picked up by CA 19-9.
Comments 4:
The discussion is too long, so it would be better to focus on clearly showing the differences from previous large-scale studies.
Response 4:
Thank you for your comment. We have organized a description of the discussion session.
Reviewer 3 Report
Comments and Suggestions for Authors
The study aimed to evaluate the clinical application of the APOA2-i index for pancreatic cancer diagnosis. The authors analyzed the diagnostic performance of blood apolipoprotein A2 isoforms (APOA2-i) in comparison with CA 19-9 for pancreatic cancer patients. They concluded that for the diagnosis of pancreatic cancer (PC), APOA2-i levels provide better prediction in the early stage of PC while CA 19-9 in the advanced stage of PC. Early diagnosis has been challenging for PC. The findings in this paper may provide some guidance for clinical application of blood test of APOA2 isoforms alone or in combination with CA 19-9 for the diagnosis of PC. The paper will be improved by addressing following concerns.
- It is better to describe how the sensitivity, specificity, positive predictive value (PPV) and negative predictive value (NPP) were measured or calculated in the Methods section. So that the results presented in Table 3 and Fig.1 and Fig.2 would be easy to understand and follow.
- Receiver operating characteristic (ROC) curves were generated for the diagnosis rates described in Section 2.3, and ROC produced an area under the curve (AUC) as described in Section 3.2 and 3.3. What are the biological meanings and clinical relevance of AUC values? What is the connection between ROC and AUC and the diagnosis rates?
- In the Discussion Section, lines 242-245, the test results of APOA2-i and CA 19-9 were compared. It is unclear what did those rates represent without referencing back to the Result section.
- About the isoforms of APOA2, three isoforms: APOA2-TQ/TQ, -TO/AT, and -AT/AT were described in the Discussion section while APOA2-i ATQ was written in the Results section. Should it be TQ not ATQ, or there is the fourth APOA2 isoform?
- Can APOA2-i isoforms be detected in saliva or urine?
The English could be improved to more clearly express the research.
Author Response
Comments 1:
It is better to describe how the sensitivity, specificity, positive predictive value (PPV) and negative predictive value (NPP) were measured or calculated in the Methods section. So that the results presented in Table 3 and Fig.1 and Fig.2 would be easy to understand and follow.
Response 1:
Thank you for your comment. I forgot to write those definition. I added in the text.
Comments 2:
Receiver operating characteristic (ROC) curves were generated for the diagnosis rates described in Section 2.3, and ROC produced an area under the curve (AUC) as described in Section 3.2 and 3.3. What are the biological meanings and clinical relevance of AUC values? What is the connection between ROC and AUC and the diagnosis rates?
Response 2:
Thank you for your comment. Generally, the higher the AUC, the higher the diagnostic rate. In this study, the novel biomarker is considered to have a relatively good diagnostic performance in the diagnosis of pancreatic cancer with an AUC of 0.697, but we considered that CA 19-9, an existing biomarker, is still more useful as a single marker for the diagnosis of pancreatic cancer. We considered that the novel biomarker is effective in combination with CA 19-9 because it picks up early exocrine dysfunction.
Comments 3: In the Discussion Section, lines 242-245, the test results of APOA2-i and CA 19-9 were compared. It is unclear what did those rates represent without referencing back to the Result section.
Response 3:
Thank you for your comment. I have reworded the wording to be clear.
Comments 4: About the isoforms of APOA2, three isoforms: APOA2-TQ/TQ, -TO/AT, and -AT/AT were described in the Discussion section while APOA2-i ATQ was written in the Results section. Should it be TQ not ATQ, or there is the fourth APOA2 isoform?
Response 4:
Thank you for your comment. ATQ is true. APOA2 is consist of five isoforms. ATQ is present, but definition was not accurate. The ratio of AT to TQ is accurate and the definition section has been corrected. (p4 l102-105)
Comments 5:
Can APOA2-i isoforms be detected in saliva or urine?
Response 5:
Thank you for your comment. Now it was not detected in saliva and urine.
Round 2
Reviewer 2 Report
Comments and Suggestions for Authors
The authors revised the manuscript well.
Author Response
Comments1: The authors revised the manuscript well.
Response1: Thank you for your comments.
Reviewer 3 Report
Comments and Suggestions for Authors
Comments on revision 1
Comments 1:
It is better to describe how the sensitivity, specificity, positive predictive value (PPV) and negative predictive value (NPP) were measured or calculated in the Methods section. So that the results presented in Table 3 and Fig.1 and Fig.2 would be easy to understand and follow.
Response 1:
Thank you for your comment. I forgot to write those definition. I added in the text. √
Comments 2:
Receiver operating characteristic (ROC) curves were generated for the diagnosis rates described in Section 2.3, and ROC produced an area under the curve (AUC) as described in Section 3.2 and 3.3. What are the biological meanings and clinical relevance of AUC values? What is the connection between ROC and AUC and the diagnosis rates?
Response 2:
Thank you for your comment. Generally, the higher the AUC, the higher the diagnostic rate. In this study, the novel biomarker is considered to have a relatively good diagnostic performance in the diagnosis of pancreatic cancer with an AUC of 0.697, but we considered that CA 19-9, an existing biomarker, is still more useful as a single marker for the diagnosis of pancreatic cancer. We considered that the novel biomarker is effective in combination with CA 19-9 because it picks up early exocrine dysfunction.√
Comments 3: In the Discussion Section, lines 242-245, the test results of APOA2-i and CA 19-9 were compared. It is unclear what did those rates represent without referencing back to the Result section.
Response 3:
Thank you for your comment. I have reworded the wording to be clear.
It is not clear what did the 4.9% and 16.6% positive rates represent, APOA2-I or diagnostic rate? How was diagnostic rate defined and calculated here?
Comments 4: About the isoforms of APOA2, three isoforms: APOA2-TQ/TQ, -TO/AT, and -AT/AT were described in the Discussion section while APOA2-i ATQ was written in the Results section. Should it be TQ not ATQ, or there is the fourth APOA2 isoform?
Response 4:
Thank you for your comment. ATQ is true. APOA2 is consist of five isoforms. ATQ is present, but definition was not accurate. The ratio of AT to TQ is accurate and the definition section has been corrected. (p4 l102-105)√
Comments 5:
Can APOA2-i isoforms be detected in saliva or urine?
Response 5:
Thank you for your comment. Now it was not detected in saliva and urine.√
Comments on the Quality of English Language
English writing could be improved to more clearly expression of the work.
Author Response
Comments 1: It is not clear what did the 4.9% and 16.6% positive rates represent, APOA2-I
or diagnostic rate? How was diagnostic rate defined and calculated here?*
Response 1: Thank you for your comment. As your comment, it is not clear.
Positive rate of APOA2-i Index were 4.9% and 16.6% each. We added in the manuscript.